# Probing active sites for carbon oxides hydrogenation on Cu/TiO$_2$ using infrared spectroscopy

Ehab Shaaban [1] & Gonghu Li[1✉]

The valorization of carbon oxides on metal/metal oxide catalysts has been extensively investigated because of its ecological and economical relevance. However, the ambiguity surrounding the active sites in such catalysts hampers their rational development. Here, in situ infrared spectroscopy in combination with isotope labeling revealed that CO molecules adsorbed on Ti$^{3+}$ and Cu$^+$ interfacial sites in Cu/TiO$_2$ gave two disparate carbonyl peaks. Monitoring each of these peaks under various conditions enabled tracking the adsorption of CO, CO$_2$, H$_2$, and H$_2$O molecules on the surface. At room temperature, CO was initially adsorbed on the oxygen vacancies to produce a high frequency CO peak, Ti$^{3+}$−CO. Competitive adsorption of water molecules on the oxygen vacancies eventually promoted CO migration to copper sites to produce a low-frequency CO peak. In comparison, the presence of gaseous CO$_2$ inhibits such migration by competitive adsorption on the copper sites. At temperatures necessary to drive CO$_2$ and CO hydrogenation reactions, oxygen vacancies can still bind CO molecules, and H$_2$ spilled-over from copper also competed for adsorption on such sites. Our spectroscopic observations demonstrate the existence of bifunctional active sites in which the metal sites catalyze CO$_2$ dissociation whereas oxygen vacancies bind and activate CO molecules.

[1] Department of Chemistry, University of New Hampshire, Durham, NH 03824, USA. ✉email: gonghu.li@unh.edu

The hydrogenation of carbon oxides ($CO_x$, including $CO_2$ and CO) to various useful products, such as fuels, has been extensively studied on the surface of various heterogeneous catalysts, to solve both environmental and energy problems[1,2]. Copper-based catalysts, for instance, demonstrated the ability to efficiently catalyze such hydrogenation reactions. Some of these catalysts have been already implemented in industry, as in the reverse water-gas shift reaction and methanol synthesis[2,3]. Despite the extensive research, there are substantial uncertainties on the mechanism and the role of active sites in $CO_x$ hydrogenation that hamper the rational development of such catalysts. Proposed mechanisms for these reactions are either of dissociative nature, in which $CO_x$ species partially or totally lose oxygen then hydrogenate to products, or of associative nature, in which hydrogen atoms bind to $CO_x$ to form various intermediate species[4–11]. Heated debate arises regarding (i) active sites that mediate these reaction steps and (ii) possible reasons behind the synergetic effects between the metal and the support. To account for the synergetic effects, for instance, formation of more active sites at metal/metal oxide interface, such as metals alloys[4,5], oxygen vacancies[12,13], and interfacial Lewis acidic sites[14–16], were discussed. Other groups, however, proposed bifunctional mechanisms[17], in which metal and support coordinate tasks in the reaction.

As a "flagship" of the reducible supports since the strong metal-support interaction (SMSI) was reported, $TiO_2$ attracted extensive research to explore its role in catalysis[18,19]. A common feature in a part of this research is that $TiO_2$ support significantly enhances the activity and the selectivity of metal catalysts in reactions that involve carbon monoxide, either as a reactant (in CO hydrogenation)[16,20,21] or as an intermediate (in $CO_2$ hydrogenation)[22–24]. Mechanistic investigations pioneered by Somorjai and co-workers highlighted the role of Lewis acidic interfacial sites, generated at the metal–titania interface, in facilitating C−O bond dissociation during carbon oxides transformation[15,16]. These interfacial sites are produced from oxygen transfer from the metal oxides to the metal sites[25–30] and were reported in different catalytic systems such as in $Pt/CeO_2$ ($Pt^+$ and $Ce^{3+}$)[28], $Pt/TiO_2$ ($Pt^+$ and $Ti^{3+}$)[29], $Cu/CeO_2$ ($Cu^+$ and $Ce^{3+}$)[30], and $Cu/TiO_2$ ($Cu^+$ and $Ti^{3+}$)[31].

In the $CO_x$ hydrogenation reactions, CO is not only a reactant or intermediate but also a probe[32,33] for distinguishing binding sites when combined with in situ infrared spectroscopy. This provides a unique opportunity to deduce information on the reaction mechanism and active sites. The ability of CO to identify surface sites and to assess their activity was widely implemented in CO oxidation, to develop catalysts for automotive emissions control[34–36], however, it is less explored in $CO_x$ hydrogenation. In the current study, in situ diffuse reflectance infrared Fourier transform spectroscopy (DRIFTS) was employed to follow CO molecules as they bind to two disparate surface sites on $Cu/TiO_2$ at room temperature. The behavior of both sites was tracked, via monitoring carbonyl peak intensity and position, as isotopically labeled $^{13}CO_2$ molecules dissociate and $H_2O$ is introduced, and as conditions are changed to in situ hydrogenation conditions. Examining the disparities in their behavior under these conditions revealed novel information that dictated an assignment different from what was proposed previously in the literature. Such observations and discussion provide insights on the bifunctional role played by the metal sites and the interface during $CO_x$ hydrogenation reaction.

## Result and discussion

### Characterization of $Cu/TiO_2$ catalysts.

Highly dispersed copper sites were prepared on a $TiO_2$ surface following our published

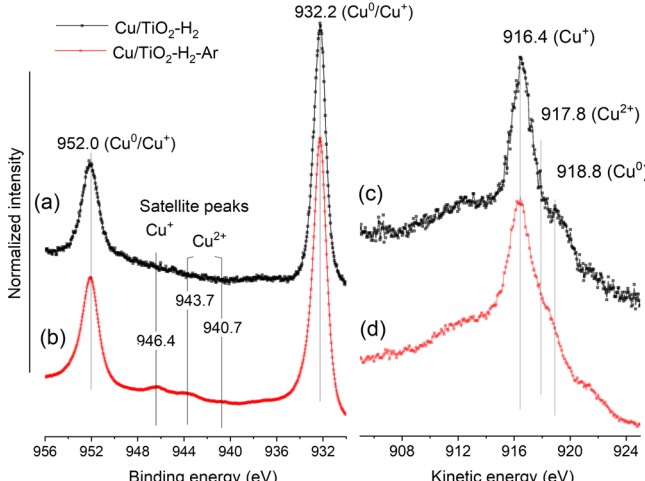

**Fig. 1 Copper X-ray photoelectron and X-ray-excited Auger electron spectra.** Cu 2p X-ray photoelectron spectra (**a**, **b**) and Cu LMM X-ray-excited Auger electron spectra (**c**, **d**) of: **a**, **c** $Cu/TiO_2$–$H_2$, after the catalyst was pretreated at 300 °C under hydrogen for 1 h; **b**, **d** $Cu/TiO_2$–$H_2$–Ar, hydrogen reduced sample was further treated at 300 °C under Ar.

procedure[37], as described in the methods section. The presence of copper sites was confirmed with X-ray photoelectron spectroscopy (XPS), CO adsorption, and UV–vis spectroscopy.

The chemical states of copper sites on the $Cu/TiO_2$ sample were investigated by examining the Cu 2p and Cu LMM regions in the X-ray photoelectron and X-ray-excited Auger electron spectra, respectively (Fig. 1). The catalyst was examined after hydrogen pre-treatment at 300 °C for 1 h (sample denoted as $Cu/TiO_2$–$H_2$), and after the reduced sample was annealed under Ar flow at 300 °C for 1 h (sample denoted as $Cu/TiO_2$–$H_2$–Ar). The XPS spectrum in the Cu 2p region indicates that $Cu^0$ and $Cu^+$ are the main species in the hydrogen-treated sample (Fig. 1a). This is evident from the presence of $Cu^0$ and/or $Cu^+$ peaks at 932.2 eV (Cu $2p_{3/2}$) and 952.0 eV (Cu $2p_{1/2}$), and the absence of $Cu^{2+}$ satellite peaks that typically emerge in between those two peaks. It is difficult to distinguish between $Cu^0$ and $Cu^+$ species based only on the XPS spectrum in the Cu 2p region[38]. Nonetheless, the Cu LMM region showed the characteristic peak for $Cu^0$ at 918.8 eV (Fig. 1c; in kinetic energy)[39,40], and the amount of $Cu^0$ was estimated to be around 37% from spectral fitting (Fig. S1).

Annealing the sample at 300 °C under Ar caused the copper speciation to shift more toward the cationic species, as the $Cu^{2+}$ and $Cu^+$ satellite peaks[38,40] became more discernable (Fig. 1b). $Cu^{2+}$ was also detected in the Cu LMM region at 917.8 eV[39], along with $Cu^+$ and $Cu^0$ (Fig. 1d). Peak deconvolutions indicate that $Cu^{2+}$ is a minor component with around 10% peak area (Fig. S1), which is consistent with the small $Cu^{2+}$ satellite peak in the XPS spectra. Note that in pure CuO, the intensity of the satellite peak is typically ~0.5 of the main Cu $2p_{3/2}$ peak[40]. The increase in the amount of oxidized species after annealing under Ar flow indicates that the $TiO_2$ support is likely reduced with copper metal to produce oxygen vacancies, as will be discussed in detail later.

To further probe the different copper sites, CO adsorption was conducted since the IR signals of surface carbonyls strongly depend on the oxidation states of the metal. The hydrogen-treated $Cu/TiO_2$ sample was loaded in the in situ diffuse reflectance cell in air and purged with CO for 15 min at room temperature. Subsequently, gaseous CO was purged by flowing Ar prior to spectrum collection. Two carbonyl peaks at 2106 and 2058 cm$^{-1}$ are present in the spectrum (Fig. S2) for CO adsorbed

on surface $Cu^+$ and $Cu^0$ sites[37,41], respectively, whereas on pure $TiO_2$ no strong carbonyl bands were observed under the same conditions. Diffuse reflectance UV–Vis spectra were also collected for the $Cu/TiO_2$ sample and pure $TiO_2$ (Fig. S3). Unlike pure $TiO_2$ that shows adsorption only in the UV region, the $Cu/TiO_2$ sample possesses absorption in the visible region between 400 and 500 nm, due to $Ti^{IV}-O-Cu^I$ metal-to-metal charge-transfer[42], and ~600–800 nm for the d-d transition of $Cu^{2+}$[,37].

**Binding sites and source of the unprompted CO.** In order to probe surface sites responsible for $CO_x$ binding, the $Cu/TiO_2$ sample was pretreated in a Harrick Praying Mantis IR cell at various temperatures (100–400 °C) under constant Ar flow. After the sample was cooled down to room temperature under Ar, the IR cell was closed and the $Cu/TiO_2$ surface was monitored with in situ DRIFTS. Despite the repeated washing for $TiO_2$ with $H_2O_2$ (see the methods section), there was always a slow and spontaneous CO formation on the surface of the pretreated $Cu/TiO_2$ sample at room temperature, as indicated by the carbonyl peaks associated with surface-adsorbed CO (Fig. 2). The CO molecules could be produced from adventitious carbon on the surface of $Cu/TiO_2$[43]. However, in our study, it is likely produced from the recombination of surface oxygen and carbon species that have been formed during the pretreatment step. Heating the sample in this step should trigger decomposition of carbonate-like species on $TiO_2$[44,45], to produce carbon oxides which in turn dissociate[46–52] on copper to form surface adsorbed oxygen and carbon, as suggested by the change in initial carbonate regions when the pretreatment temperature was increased, Fig. 2.

At low temperatures pretreatments, 100 and 150 °C, a small fraction of adsorbed water was removed and only a low frequency (LF) CO peak was observed on $Cu/TiO_2$, Fig. 2a, b. On $Cu/TiO_2$ pretreated at 200 °C and higher, however, more water was

removed from the surface and two distinct CO peaks were observed, the LF peak and another CO peak located at a higher frequency (HF), (Fig. 2c–e). Interestingly, the initial peak positions for both the HF and LF peaks showed a strong dependence on pretreatment temperatures. The onset peak position for both peaks is blue-shifted with the increase in the pretreatment temperature. Furthermore, with time after a given pretreatment, the HF peak initially increased in intensity and then decayed at the same wavenumber, whereas the LF peak appeared later and underwent similar changes in peak intensity but red-shifted until it fully decayed. For instance, in the sample pretreated at 300 °C, the HF peak appeared at ~2130 cm$^{-1}$ ($\nu_{CO}$). The intensity of this peak increased gradually and then decreased while the LF peak started to emerge at ~2118 cm$^{-1}$ (Fig. 2d). The LF peak slowly shifted to 2111 cm$^{-1}$ before its disappearance.

Both CO peaks showed different sensitivity to the residual adsorbed water that remained on the surface after different temperature pretreatments. Comparing the IR regions of surface adsorbed water, either molecularly adsorbed at ~1620 cm$^{-1}$ (Fig. 2f–j) or dissociatively adsorbed as hydroxyls at 3700–3000 cm$^{-1}$ (Fig. S4)[53,54] indicates that the HF CO peak emerges at lower water coverage compared to the LF peak, as can be seen from the initial spectra in h–j in comparison to f and g in Fig. 2. This difference in sensitivity toward the residual water as a Lewis base indicates that the HF site is more Lewis acidic than the LF site. Furthermore, water eventually re-accumulated on the sample surface with the extended time of data collection (up to a few hours in some experiments). The source of this water is the trace amount of adsorbed water that usually exists on the cold inner surfaces of the sample cell walls, a common issue in DRIFTS and other surface studies[54,55]. In each experiment, the re-adsorption of water on the sample (Fig. 2h–j) was accompanied by the movement of the CO from the HF to LF sites (Fig. 2c–e). This indicates that the re-adsorbed water will eventually replace CO, since $H_2O$ is a stronger Lewis base, on the HF site prompting CO migration to the less acidic LF site. More discussion is presented in "Water adsorption" section.

The above observations indicate the presence of two disparate CO adsorption sites (HF and LF sites) on the $Cu/TiO_2$ surface. For metal catalysts supported on reducible metal oxides, it's well documented that thermal treatment under an oxygen-deficient atmosphere triggers oxygen transfer from the metal oxides to the metal sites at the interfacial region[25–30]. Such interactions generate acidic interfacial sites as demonstrated in $Pt/CeO_2$ ($Pt^+$ and $Ce^{3+}$)[28], $Pt/TiO_2$ ($Pt^+$ and $Ti^{3+}$)[29], and $Cu/CeO_2$ ($Cu^+$ and $Ce^{3+}$)[30]. Similarly, thermal treatment of $Cu/TiO_2$ samples under inert gas flow leads to the formation of $Cu^+$ and $Ti^{3+}$ sites at the $Cu/TiO_2$ interface[31]. Such Lewis acidic sites at the interfaces can bind to Lewis basic molecules such as $H_2O$ and CO. In the case of CO, the metal cations that possess partially filled d-shell can back-bond to CO, resulting in IR stretch bands lower than 2143 cm$^{-1}$, where the IR band of gaseous CO locates[56]. In accordance with the above discussion, the LF CO peak (2118–2111 cm$^{-1}$) observed in the spectra shown in Fig. 2 can be attributed to the carbonyl stretching mode of CO adsorbed on the surface $Cu^+$ sites[31,43]. The adsorption site corresponding to the HF CO band (e.g., 2131 cm$^{-1}$ on $Cu/TiO_2$ pretreated at 400 °C), however, is more acidic since it required higher pre-treatment temperatures and was more sensitive to water adsorption. More importantly, no progressive redshift was observed in the HF CO peak over time, suggesting that it is associated with a single adsorption site. For these reasons and based on previous theoretical[57] and experimental[58–62] studies, the observed HF peak can be assigned to CO adsorbed on oxygen vacancies ($Ti^{3+}$). Rigorous pre-treatment conditions (e.g., prolonged treatment at 450 °C under ultra-high vacuum)

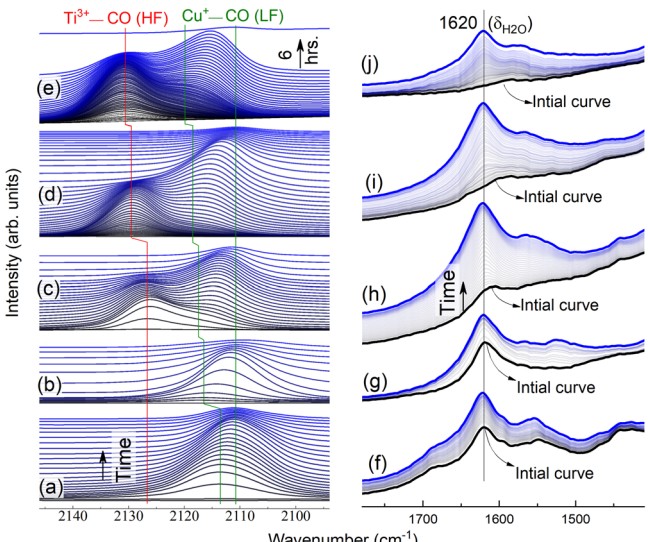

**Fig. 2 DRIFTS spectra of $Cu/TiO_2$ after activation at different temperatures.** Carbonyl (**a–e**) and the corresponding carbonate (**f–j**) regions of $Cu/TiO_2$ as a function of time (10 min between spectra) after pre-treatment under Ar flow at (**a, f**) 100 °C, (**b, g**) 150 °C, (**c, h**) 200 °C, (**d, i**) 300 °C, and (**e, j**) 400 °C. Spectra were collected after the samples are cooled to room temperature. The carbonyl region demonstrates the change in intensity and position of two distinct CO binding sites, low frequency (LF) and high frequency (HF), on $Cu^+$ and $Ti^{3+}$, respectively. The corresponding carbonate region indicates the surface adsorbed water at 1620 cm$^{-1}$.

were often needed to create such CO adsorption sites on pure TiO$_2$ surfaces[58–62]. In the present study, thermal treatment at 200 °C under Ar was sufficient to create oxygen vacancies on TiO$_2$, due to the presence of surface Cu sites which facilitates the formation of such sites. It is worth mentioning that multiple studies demonstrated that metals supported on reducible metal oxides facilitates the formation of oxygen vacancies which get stabilized via metal/metal oxide Schottky junction[27,63,64].

**Isotope studies using $^{13}CO_2$.** To probe the roles of oxygen vacancies and surface Cu$^+$ sites in CO$_2$ dissociation, we carried out isotope labeling experiments where different amounts of $^{13}CO_2$ were introduced into the IR cell after Cu/TiO$_2$ was thermally treated at 300 °C and cooled down to room temperature under Ar. Formation of both CO isotopes, $^{12}CO$ from surface carbon residues and $^{13}CO$ from gaseous $^{13}CO_2$, on the HF/LF sites was monitored as a function of time and as the amount of $^{13}CO_2$ admitted was increased (Fig. 3).

The presence of a relatively small amount of $^{13}CO_2$ (0.1 bar) led to the formation of $^{13}CO$ on Cu/TiO$_2$, as indicated by the appearance of the HF peak at 2082 cm$^{-1}$ (Fig. 3b). The evolution of this peak follows the same pattern as the $^{12}CO$ HF peak at 2130 cm$^{-1}$, which gradually decayed while the LF peak started to develop. Both CO isotopes followed almost identical behavior in evolution, in terms of preferential adsorption on the HF sites and their migration to the LF sites. Increasing the pressure of $^{13}CO_2$ to 1 bar resulted in more surface-adsorbed $^{13}CO$ (Fig. 3c). In the spectra collected immediately after the introduction of $^{13}CO_2$, the intensity of the $^{13}CO$ HF peak at 2082 cm$^{-1}$ is significantly greater than that of the $^{12}CO$ HF peak. This is likely because the amount of $^{13}CO$ produced from gaseous $^{13}CO_2$ is much larger than that of $^{12}CO$ produced from surface carbon. The increase in the onset relative amount of $^{13}CO$ as the amount of injected $^{13}CO_2$ increased confirms the occurring of dissociation, whereas the relentless formation of $^{12}CO$ and its displacement to $^{13}CO$ from their binding sites supports that each of the CO isotopes originates from two opposing reactions on the catalyst as discussed earlier.

The presence of gaseous $^{13}CO_2$ inhibited the migration of CO adsorbed on oxygen vacancies (the HF peak) to surface Cu$^+$ sites (the LF peak), as shown by the comparison in Fig. 3. In the absence of gaseous $^{13}CO_2$ (Fig. 3a), the LF peak shifted from 2118

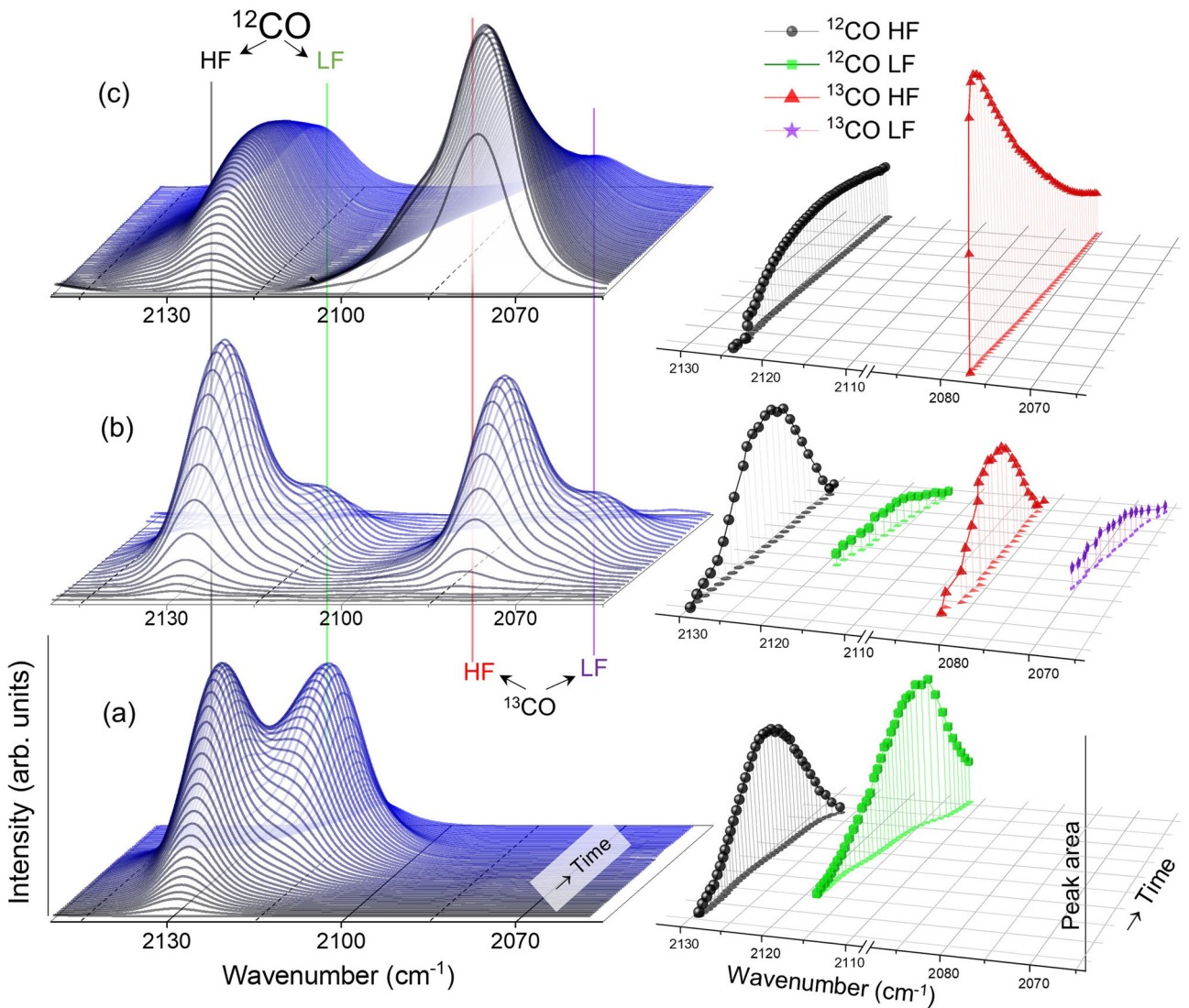

**Fig. 3 $^{13}CO_2$ isotope labeling experiment.** DRIFTS spectra of surface-adsorbed CO as a function of time on Cu/TiO$_2$ samples pretreated at 300 °C under Ar flow. Different amounts of $^{13}CO_2$ were present in the IR cell: **a** 0 bar, **b** 0.1 bar, and **c** 1 bar. To the right, the corresponding integrated CO peak areas are plotted as a function of wavenumber and time.

to 2111 cm$^{-1}$, and its maximum integrated area was slightly greater than that of the HF peak. Introducing a small amount of $^{13}CO_2$ (0.1 bar, Fig. 3b) significantly reduced the amount of CO adsorbed on the surface Cu$^+$ sites, as shown by the relatively small integrated areas of the LF peaks (both $^{12}CO$ and $^{13}CO$). Further increasing the amount of gaseous $^{13}CO_2$ led to a nearly complete absence of the LF peak and slow decay of the HF peak (Fig. 3c). These results suggest that the gaseous $^{13}CO_2$ competes with both CO isotopes during adsorption on the Cu$^+$ sites but not on the Ti$^{3+}$ sites. This is further supported by previous studies demonstrating that $CO_2$ became strongly adsorbed as carbonates on oxidized copper sites[46,65,66].

**Water adsorption**. As an omnipresent Lewis base, water molecules bind to the interfacial acidic sites on the surface of the Cu/TiO$_2$ catalyst. Since Ti$^{3+}$ is a stronger Lewis acid than Cu$^+$ and H$_2$O is a stronger Lewis base than CO, the activation temperature required to desorb water from the Ti$^{3+}$ was higher and CO was preferably adsorbed on such site compared to Cu$^+$. On Cu/TiO$_2$ samples with adsorbed CO, readsorption of water will replace any adsorbed CO on the Ti$^{3+}$ sites then Cu$^+$ sites, as discussed in "Binding sites and source of the unprompted CO" section.

The role of water was further confirmed by purposely introducing water vapor in the middle of the rise of the HF CO peak. Spectra were collected for a Cu/TiO$_2$ sample at room temperature under continuous Ar flow after pretreatment at 300 °C. During the rise of the HF, Ar flow was bypassed to flow above degassed water for 3 s then the Ar was switched back to the dry flow. As can be seen in Fig. 4, the decay of the HF peak and the rise/decay of the LF peak occurred immediately after water introduction, in less than two minutes as compared to a few hours in the absence of dosed water. The decay in the CO peaks was accompanied also by a rapid rise in the molecularly adsorbed water at 1620 cm$^{-1}$ and hydroxyl peak at 3695 cm$^{-1}$ (Fig. 4a, c). The latter peak is attributed to water healing oxygen vacancies in the vicinity of copper sites. Multiple studies[54,67,68] have shown that this hydroxyl peak (peak at 3695 cm$^{-1}$) is for Ti$^{4+}$-OH peaks that was previously reactive defect site (Ti$^{3+}$) but was filled with dissociatively adsorbed water.

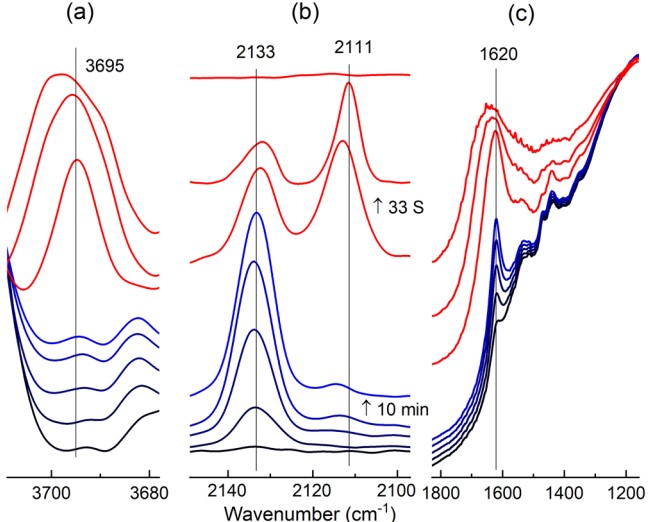

**Fig. 4 DRIFTS spectra upon introduction of water vapor.** Changes in the **a** hydroxyl, **b** carbonyl, and **c** water regions in the DRIFTS spectra of Cu/TiO$_2$ before (blue, collected every 10 min) and after (red, collected every 33 s) the introduction of water vapor into the IR cell.

**H$_2$ admission**. In our study, introducing hydrogen at room temperature to activated Cu/TiO$_2$ either before or during the rise of CO on the HF site did not interrupt the behavior of surface adsorbed CO, either during the rise of CO on the HF site or its transition to the LF site. Such behavior could be attributed to the inactivity of copper toward hydrogen at room temperature[69,70].

After pretreatment at 300 °C under H$_2$ flow, only the LF CO peak was observed in the DRIFTS spectra upon CO injection. However, upon a second pretreatment under Ar, both HF and LF CO peaks appeared (spectra not shown here for brevity). Likely, hydrogenic species formed on the HF binding site upon hydrogen pretreatment is responsible for the absence of the HF CO peak. In line with these observations, CO adsorption on Cu/TiO$_2$ studies showed that the appearance of the HF peak required the application of delicate pretreatment conditions[71,72], however, in such studies the HF site was assigned to different binding sites. Thermal pretreatment with hydrogen precluded the formation of the HF peak, and re-oxidation with N$_2$O could not retrieve it[71]. When the H$_2$ pretreatment was followed with an evacuation step at the same reduction temperature, the HF peak was observed with even an enhanced intensity[71,72]. This intriguing behavior could be attributed to the competitive adsorption of CO and H$_2$ on surface oxygen vacancies. This behavior was demonstrated previously, with different characterization tools, over Pt/TiO$_2$[29] and Rh/TiO$_2$[73–76] catalysts. In such studies, it was concluded that the metal site facilitates the reduction of the titania support to produce hydrogenic species, which inhibited CO adsorption on titania.

To further investigate the role of the HF and LF CO binding sites in carbon oxides hydrogenation, those two peaks were monitored as the H$_2$ gas is co-adsorbed and as the temperature is increased to the reaction temperature. When the sample was purged with a mixture of Ar and CO at room temperature, only the LF CO peak was observed (Fig. 5a, b). As the temperature increased, the CO peak shifted gradually from the LF sites to the HF sites. Moreover, when Ar flow in the gas mixture was replaced with an equal amount of hydrogen, the HF CO adsorption peak was not affected until the temperature reached 200 °C. At such temperature (and higher), a red-shift and a rapid decrease in the peak intensity were observed as H$_2$ was admitted (Fig. 5c), for H$_2$ and CO coadsorption at 275 °C. Starting at 275 °C and above, the production of methane gas was observed. The CO peak intensity and position were partially retrieved again when H$_2$ and CO flow was switched back to Ar and CO flow. Such cyclic changes in HF peak occurred whenever the flow was switched back and forth between Ar + CO and H$_2$ + CO. The observed ability of the HF site (Ti$^{3+}$) to maintain CO binding at high temperatures and to interact with both H$_2$ and CO at different temperatures strongly suggests that such sites are the active centers in CO hydrogenation over Cu/TiO$_2$.

**CO$_2$ dissociation and carbon oxides chemical conversion**. The observed disparities between the surface binding sites confirm the existence of bi-functional catalytic sites that can work collaboratively to catalyze CO$_2$ hydrogenation. Such bifunctionality enables heterogeneous catalysts to efficiently catalyze CO$_2$ conversion to higher hydrogenated products with one-pot synthesis[77,78]. As discussed earlier, the introduction and dissociation of CO$_2$ affected only the LF site and the produced CO accumulated on the HF sites (Fig. 6). Such observations confirm the role of the metal sites in the dissociative adsorption of CO$_2$. Multiple studies have demonstrated that CO$_2$ dissociation, a key step in CO$_2$ chemical conversion, takes place spontaneously on pure copper metal[46–50]. It is worth mentioning that accumulation of the produced oxygen poisons the copper metal surface[50,79],

**Fig. 5 DRIFTS spectra for high-temperature CO and CO-H$_2$ adsorption. a** CO adsorption peaks under continuous CO flow over Cu/TiO$_2$ and as the temperature increases. The spectra were collected 10 min. after reaching the desired temperature. **b** Peak areas for the HF and LF CO peaks that were deconvoluted from **a**. **c** CO adsorption peaks at 275 °C under continuous CO flow over Cu/TiO$_2$, and as the flow was switched back and forth between Ar + CO and H$_2$ + CO.

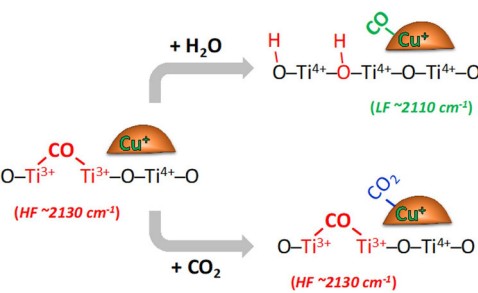

**Fig. 6 Schematic of competitive adsorption of CO with H$_2$O and CO$_2$.** H$_2$O competes with CO on oxygen vacancies while CO$_2$ adsorbs on copper sites.

which can be regenerated with hydrogen, based on redox mechanism[80] in reverse water gas shift reaction. Moreover, studies of CO$_x$ hydrogenation have demonstrated that surface copper sites can protect oxygen vacancies, the proposed active sites, from healing when CO$_2$ is introduced to a feed mixture of CO and H$_2$[12].

The HF site demonstrated a higher affinity to bind CO even at high temperatures. This suggests that the oxygen vacancies will stabilize the CO produced from CO$_2$ dissociation occurring on the neighboring metal sites. And since hydrogenic species

compete with CO on such sites upon H$_2$ introduction, it can be concluded that oxygen vacancies activate both molecules for the reaction. Calculations and experimental observation in multiple studies have confirmed such role of the oxygen vacancies during the conversion of syngas (or a mixture of syngas and CO$_2$) to methanol on reducible metal oxides and copper supported on reducible metal oxides[12,13,63,81,82].

The observed strong adsorption of water molecules on the acidic interfacial sites, which displace CO from HF binding sites, highlights the detrimental role of water on the interfacial active sites during the hydrogenation reaction. This is in line with the previous observations that water produced during CO$_2$ hydrogenation deactivates the Cu/metal oxide catalysts, and for this reason, CAMERE (Carbon dioxide hydrogenation to form methanol via a reverse-water gas shift reaction) process was implemented to minimize water percentage in the reaction mixture[11]. In such process CO$_2$ was reduced first to CO then the CO was fed to another reactor for further reaction.

## Conclusion

We have employed in situ DRIFTS to investigate surface sites responsible for CO$_x$ hydrogenation on Cu/TiO$_2$. Introducing $^{13}$CO$_2$ at room temperature to a thermally activated Cu/TiO$_2$ catalyst produces a mixture of $^{13}$CO and $^{12}$CO that likely originated

from $^{13}CO_2$ spontaneous dissociation and carbon residue oxidation, respectively. The ratio of $^{13}CO/^{12}CO$ isotopes increased as the introduced amount of $^{13}CO_2$ was increased, however, with time $^{12}CO$ eventually displaced $^{13}CO$ on the catalyst surface.

Interfacial sites in $Cu/TiO_2$ catalyst gave rise to two distinct CO binding sites, $Cu^+$ and $Ti^{3+}$. The $Cu^+$-CO spanned the range 2118–2111 cm$^{-1}$ whereases the $Ti^{3+}$-CO (CO on oxygen vacancies) gave rise to a carbonyl peak at a single wavenumber in the range 2126–2131 cm$^{-1}$, depending on the activation temperature employed. Being more acidic, the HF site ($Ti^{3+}$) required a higher temperature (200 °C or higher) to relinquish adsorbed atmospheric water and showed more affinity to CO than the LF site ($Cu^+$). Likewise, water re-adsorption on the surface that already contains adsorbed CO prompted the migration of CO from the oxygen vacancies to the neighboring $Cu^+$ sites. $CO_2$ admission, on the other hand, suppressed the LF CO peak area and limited the CO migration from the oxygen vacancies.

Hydrogenic species, formed from $H_2$ spill-over during pretreatment, prevented the formation of the CO HF peak at room temperature. However, this did not prevent the formation of the LF CO peak. Furthermore, the HF sites demonstrated the ability to interact with both CO and $H_2$ at high temperatures necessary to form methane.

On the $Cu/TiO_2$ surface, the adsorption of CO was affected by the presence of other molecules in the hydrogenation reaction mixture, including $CO_2$, CO, $H_2$, and $H_2O$. The observed disparities of carbonyl signals suggest the existence of bifunctional catalytic sites, in which metallic copper sites serve as $CO_2$ dissociation sites, whereas the $Cu^+$ and the oxygen vacancies bind the produced CO molecules for further reductions.

## Methods

The $Cu/TiO_2$ catalyst was prepared via a simple precipitation method using $CuCl_2$ (Sigma-Aldrich, 99.995%), P25 ($TiO_2$, obtained from Evonik), and ammonia (aq, BDH, ACS grade). To remove surface organic contaminants, 300 mg $TiO_2$ was washed with 30 mL of 1:3 by volume solution of 30% $H_2O_2$ (aq, J.T. Baker, CMOS grade) in Milli-Q water, using sonication to disperse the powder and centrifugation to retrieve it. A 10 mL solution of ammonia was added to 30 mL of the washed $TiO_2$ suspension before adding 10 mg of $CuCl_2$ under constant stirring. The resulting $Cu/TiO_2$ was separated by centrifugation, washed with Milli-Q water, and vacuum dried at room temperature overnight. Prior to infrared studies, the synthesized $Cu/TiO_2$ was annealed under a flow of $H_2$ for 3 h at 300 °C in a tube furnace.

Additional pre-treatment of powder samples was done in a Harrick Praying Mantis diffuse reflectance IR cell attached to a Thermo Nicolet 6700 FTIR spectrometer. The catalyst in powder form was compressed in the sample holder and purged with Ar (99.999%) prior to thermal treatment at the desired temperature for 1 h under Ar flow (400 mL/min). The sample was then cooled down to room temperature under continuous Ar flow before the IR cell was closed and spectra were collected. In the isotope experiments, specific amounts of gaseous $^{13}CO_2$ were introduced into the IR cell by using a syringe or flowing $^{13}CO_2$ briefly to produce desired pressure. Water vapor was introduced into the IR cell by using a bubbler with Ar flow. Typically, IR spectra were collected every 10 min, unless otherwise mentioned. UV–Visible spectra were obtained on a Cary 50 Bio spectrophotometer. A Barrelino diffuse reflectance probe was used to collect UV–Visible spectra of powder samples using $BaSO_4$ as a standard. X-ray Photoelectron Spectroscopy (XPS) investigations were performed with a KRATOS Axis Supra. Monochromated Al Kα was used for the excitation of the photo and Auger electrons. The binding energy scale was referenced to the C1s at 285 eV. A glove bag was used to ensure that the hydrogen-treated sample is always kept under a dry atmosphere of Ar during loading in the XPS instrument.

## Data availability

Any relevant data are available from the authors upon reasonable request.

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

## Acknowledgements
This material is based upon work supported by the U.S. National Science Foundation under grants 1705528 and 2102655. The authors thank Professor N. Aaron Deskins for his insightful discussions.

## Author contributions
E.S and G.L. conceived the idea and planned the research. E.S. carried out the experiments. E.S. and G.L. analyzed the results and wrote the manuscript.

## Competing interests
The authors declare no competing interests.
