## [Peer Review File · Communications Chemistry]

Reviewers' comments:

Reviewer #1 (Remarks to the Author):

This work studied the roles of oxygen vacancies and Cu⁺ site in Cu/TiO₂ for CO₂ dissociation at 0.1-1.0 bar, and identified the active sites of CO/CO₂ hydrogenation using in situ infrared spectroscopy in combination with isotope labeling. It was concluded that the interfacial Cu⁺ and Ti³⁺ sites competitively adsorb CO discriminated by the LF (low frequency, 2130 cm⁻¹) and HF (high frequency, 2110 cm⁻¹) bands, while CO₂ and CO competitively adsorb on the Cu⁺ site but not on the Ti³⁺ site. Introduction of H₂O in the system facilitated CO adsorption on the Cu⁺ sites. HF site interacting with both CO and H₂ facilitated the formation of CH₄. These observations deepen the understanding of the adsorption and activation of small molecules on Cu⁺ and oxygen vacancies in CO₂/CO hydrogenation. But the less structural information for the hydrogen-treated sample weakened the conclusions, and therefore the paper needs to be updated adequately before it can be considered for publication.

1. General information on the examined sample is insufficient. Copper catalysts for CO/CO₂ hydrogenation are typically subjected to hydrogen reduction at mild temperatures for converting CuO into metallic or just positively-charged copper species that enhance CO adsorption. Here, the Cu/TiO₂ sample was annealed at 300 °C by flowing H₂. However, the structural characters of this hydrogen-treated sample are missing, especially sizes and chemical states of the different copper species, number of the oxygen vacancies on TiO₂, and their interfacial geometry. While the author further claimed that thermal treatment of Cu/TiO₂ under Ar flow at 200 °C led to the formation of Cu⁺ and Ti³⁺ sites at the interface. The variations in these sites during the above two stages should be clarified.
2. Discrimination Cu⁰/Cu⁺/Cu²⁺ by XPS of Cu 2p is not convincing. Since Cu⁺ played a key role in this work, in situ Cu Auger experiment should be performed to quantify this key species. Please note, metallic Cu is easily oxidized when exposed to air.
3. The introduction of H₂O facilitated CO adsorption on Cu⁺ site. This should be interpreted more clearly.
4. The hydroxyl band (OH) at 3695 cm⁻¹ was assigned to Ti⁴⁺ site; but it was also linked to Ti³⁺ site by the author. This is controversial.
5. CO₂ was proposed to mainly adsorb and dissociate on metallic Cu. The ratio of Cu⁺ and Cu⁰ should be quantified.

Reviewer #2 (Remarks to the Author):

In this work, Shaaban and Li applied in situ DRIFTS to study CO_x hydrogenation process and active sites on a Cu/TiO₂ catalyst. They applied introduced ¹³CO₂ to the system and detected ¹³CO and ¹²CO and their evolution. They identified two sites, HF sites (Ti³⁺) and LF site (Cu⁺) and explained their disparities in CO adsorption and competitive water adsorption. This work provided useful insights to the reaction mechanism in CO_x hydrogenation on Cu/TiO₂. Some revisions are required

before it can be published.

Specific comments:

1. There are several grammar mistakes in the manuscript. For example, in the first paragraph in the introduction section, it was written "... COx is partially or totally lose oxygen then hydrogenate to the products,". The authors should go over the manuscript and correct them.

2. The caption of Figure 1 should include the information regarding the peaks with binding energy of ~952-953 eV.

3. In section 2.1, the authors described the observation of CO peak shift in the range of 2140 – 2100 cm⁻¹. According to the US National Institute of Standards and Technology, the CO peak in the similar range corresponds to gaseous CO instead of bonded CO. Please check this reference and confirm if the conclusion still stands. <https://webbook.nist.gov/cgi/cbook.cgi?ID=C630080&Type=IR-SPEC&Index=1>.

4. Additionally, it is not easy to understand why water peak can increase after the IR cell has been under constant Ar purging. It is true that water molecules show a strong IR absorption at ~1630 cm⁻¹. However, did IR spectrum show a strong and wide absorption band centered at 3000+ cm⁻¹? Please again check the US National Institute of Standards and Technology for the water IR spectrum. <https://webbook.nist.gov/cgi/cbook.cgi?ID=C7732185&Type=IR-SPEC&Index=1>. As shown in Figure 5, the strong and wide band was not observed at the same time when 1620 cm⁻¹ peak was observed, then it is likely the 1620 cm⁻¹ peak is related to carbonate.

Reviewer #1 (Remarks to the Author):

This work studied the roles of oxygen vacancies and Cu^+ site in Cu/TiO_2 for CO_2 dissociation at 0.1-1.0 bar and identified the active sites of CO/CO_2 hydrogenation using in situ infrared spectroscopy in combination with isotope labeling. It was concluded that the interfacial Cu^+ and Ti^{3+} sites competitively adsorb CO discriminated by the LF (low frequency, 2130 cm^{-1}) and HF (high frequency, 2110 cm^{-1}) bands, while CO_2 and CO competitively adsorb on the Cu^+ site but not on the Ti^{3+} site. Introduction of H_2O in the system facilitated CO adsorption on the Cu^+ sites. HF site interacting with both CO and H_2 facilitated the formation of CH_4 . These observations deepen the understanding of the adsorption and activation of small molecules on Cu^+ and oxygen vacancies in CO_2/CO hydrogenation. But the less structural information for the hydrogen-treated sample weakened the conclusions, and therefore the paper needs to be updated adequately before it can be considered for publication.

Reviewer #1 remarks	Authors' Response
1. General information on the examined sample is insufficient. Copper catalysts for CO/CO_2 hydrogenation are typically subjected to hydrogen reduction at mild temperatures for converting CuO into metallic or just positively-charged copper species that enhance CO adsorption. Here, the Cu/TiO_2 sample was annealed at $300\text{ }^\circ\text{C}$ by flowing H_2 . However, the structural characters of this hydrogen-treated sample are missing, especially sizes and	We appreciate the Reviewer's insightful suggestions. A dedicated characterization section for the Cu/TiO_2 catalyst, section 2.1, was added. We have used X-ray photoelectron spectroscopy (XPS) and X-ray-excited Auger electron spectroscopy (XAES) to characterize copper oxidation. XAES results indicate that Cu^0 co-exists with Cu^+ in the hydrogen-treated sample, estimated to be 37% and 63%, respectively. Annealing the sample under Ar causes more speciation toward cationic copper, with estimated peak areas of 30%, 60%, and 10% for Cu , Cu^+ , and Cu^{2+} , respectively. This is consistent with the discussion on copper reduction on TiO_2 . We also added the DRIFTS study of CO adsorption to the Supporting Information to further support the existence of both Cu^+ and Cu^0 in the sample. Furthermore, at such low copper

chemical states of the different copper species, number of the oxygen vacancies on TiO₂, and their interfacial geometry. While the author further claimed that thermal treatment of Cu/TiO₂ under Ar flow at 200 °C led to the formation of Cu⁺ and Ti³⁺ sites at the interface. The variations in these sites during the above two stages should be clarified.	loadings, the Cu species are highly dispersed and characterization with TEM was not able to quantify the particle size, as shown in our previous study.¹ The technique that we are familiar with that can quantitatively determine the number of oxygen vacancies is XPS. And from our measurement and others² it is elusive to determine the amount of Ti³⁺ since it exists as a minority interfacial site along with the dominant Ti⁴⁺ cations in the TiO₂. Regarding the variation of Cu⁺ and Ti³⁺ site during Ar and H₂ treatment, we believe that section 2.5 (2.4 before the edit) on H₂ admission provides a discussion to understand such variation. In this section:  • We presented observations, supported with literature, that hydrogen competes with CO on Ti³⁺ site, where H₂ forms hydrogenic species that prevents CO adsorption on titania. So, H₂ pretreatment prevents the formation of the HF peak (for Ti³⁺-CO). , however, on a second treatment for the same sample under Ar, we could observe the HF peak. In both cases, the LF peak for CO on Cu⁺ was not affected. • This competitive adsorption also happened when we kept switching between Ar and H₂ as CO was continuously flowing at higher temperatures (Figure 5c). • We observed that heating under Ar was necessary to liberate Cu⁺ then Ti³⁺ from surface adsorbed water, which will accumulate on the sample once
--	---

	the sample is exposed to the atmosphere, see section 2.2.
2. Discrimination $\text{Cu}^0/\text{Cu}^+/\text{Cu}^{2+}$ by XPS of Cu 2p is not convincing. Since Cu^+ played a key role in this work, in situ Cu Auger experiment should be performed to quantify this key species. Please note, metallic Cu is easily oxidized when exposed to air.	As discussed above, we used XAES to characterize copper oxidation after different pretreatment. A dedicated section was added for this purpose. In such experiments, samples were loaded onto the instrument under Ar with the help of a glove bag (see the experimental section in SI).
3. The introduction of H_2O facilitated CO adsorption on Cu^+ site. This should be interpreted more clearly.	We revisited this and added a new paragraph on page 7, and updated Figure 2 (previously Figure 3) to include more clarification on the role of water. Also, we added a scheme (Figure 6) to further clarify this.
4. The hydroxyl band (OH) at 3695 cm^{-1} was assigned to Ti^{4+} site; but it was also linked to Ti^{3+} site by the author. This is in controversy.	We hope that the scheme shown in Figure 6 may help to clarify this. Based on our observations and observation from others (ref. no 55,68,69 on in the manuscript), this hydroxyl peak formed when water dissociates on Ti^{3+} , forming Ti^{4+} .
5. CO_2 was proposed to mainly adsorb and dissociate on metallic Cu. The ratio of Cu^+ and Cu^0 should be quantified.	As discussed above, we have used XAES to quantify different copper species. The results are described in Section 2.1 of the revised manuscript.

Reviewer #2 (Remarks to the Author):

In this work, Shaaban and Li applied in situ DRIFTS to study CO_x hydrogenation process and active sites on a Cu/TiO₂ catalyst. They applied introduced ¹³CO₂ to the system and detected ¹³CO and ¹²CO and their evolution. They identified two sites, HF sites (Ti³⁺) and LF sites (Cu⁺) and explained their disparities in CO adsorption and competitive water adsorption. This work provided useful insights to the reaction mechanism in CO_x hydrogenation on Cu/TiO₂. Some revisions are required before it can be published.

Specific comments:

Reviewer #2	Authors' Response
1. There are several grammar mistakes in the manuscript. For example, in the first paragraph in the introduction section, it was written "... CO _x is partially or totally loses oxygen then hydrogenate to the products, ...". The authors should go over the manuscript and correct them.	Thanks very much for bringing this to our attention. We went over the manuscript and corrected the grammar mistakes.
2. The caption of Figure 1 should include the information regarding the peaks with binding energy of ~952-953 eV.	We added a new section (2.1) that contains more details and discussion.

3. In section 2.1, the authors described the observation of CO peak shift in the range of 2140 – 2100 cm^{-1}. According to the US National Institute of Standards and Technology, the CO peak in the similar range corresponds to gaseous CO instead of bonded CO. Please check this reference and confirm if the conclusion still stands. https://webbook.nist.gov/cgi/cbook.cgi?ID=C630080&Type=IR-SPEC&Index=1.	In section 2.2 (2.1 in the previous version), the IR peaks in the range 2140 – 2100 cm^{-1} is for bonded CO instead of gaseous CO due to the following reasons:  The higher frequency (2175 cm^{-1}) branch for gaseous CO was not observed in any of the experiments in section 2.2, see Figure 2. Gaseous CO typically gives two equivalent and broad IR peaks centered at ~ 2175 and 2120 cm^{-1} for P- and R-branch, respectively, due to rovibrational transition, see the link provided by Reviewer #2. Once CO binds to catalytic surface sites, it gives IR peaks that are either above or below 2143 cm^{-1}, depending on the type of bond involved. We briefly referred to this in the manuscript (page 2 and page 8), and presented the references for details (for instance references 32, 33, and 57 in the manuscript). We specifically observed IR peaks for CO adsorbed on Cu^+ (2120-2110 cm^{-1}) and Ti^{3+} at 2130 cm^{-1} which has been reported by others, see the discussion on page 8 and reference therein (ref. 57 to 63). It is possible to distinguish between CO(ads.) and CO(g) even if both co-exist. Both bonded and gaseous CO were detected only when CO was introduced at high pressures, see Figure 6 (a) and Figure S1. Note that adsorbed CO peak at room temperature appears as large peak at 2110 cm^{-1} that sets on the P- and R-branch for gaseous CO. However, only adsorbed CO was observed after Ar purge (see Figure S1). Based on the above discussion, we could clearly detect adsorbed CO when it emerges from the surface during CO_2 dissociation and/or carbon residue oxidation, note that the DRIFTS technique is highly sensitive to surface adsorbed CO, detects CO at 10^{-6} of
---	---

	a monolayer. ³
4. Additionally, it is not easy to understand why water peak can increase after the IR cell has been under constant Ar purging. It is true that water molecules show a strong IR absorption at ~1630 cm⁻¹. However, did IR spectrum show a strong and wide absorption band centered at 3000+ cm⁻¹? Please again check the US National Institute of Standards and Technology for the water IR spectrum. https://webbook.nist.gov/cgi/cbook.cgi?ID=C7732185&Type=IR-SPEC&Index=1. As shown in Figure 5, the strong and wide band was not observed at the same time when 1620 cm⁻¹ peak was observed, then it is likely the 1620 cm⁻¹ peak is related to carbonate.	We appreciate the Reviewer's insights. Here is our clarification. a) Residual water usually exists on the cold inner walls of the instrument sample cell, a common issue that persists even under UHV, see ref. ⁴ As samples cool down from thermal annealing, it is reasonable that trace amounts of water will desorb from the wall and re-adsorbed on samples, especially with the extended time of the experiment. This was encountered in similar DRIFTS studies on TiO₂. ⁵ b) We did observe a broadband > 3000 cm⁻¹ for adsorbed water in the spectra, as can be seen from the full spectra shown in Figure S4 in the Supporting Information. Furthermore, there was a slow gradual increase in this region simultaneous with the increase in the 1620 cm⁻¹ peak, due to re-adsorption of residual water (see next point). Both the 3000+ and 1620 cm⁻¹ regions were greatly and simultaneously enhanced when water was introduced to the system which supports this assignment. c) In Figure 4 (previously Figure 5), we presented only the 3695 cm⁻¹ peak since it emerged later compared to the other hydroxyls in the 3000+ cm⁻¹ region. These latter hydroxyls simultaneously emerged with the 1620 cm⁻¹ peak, as seen in the full spectra shown in Figure S4. The IR spectrum for water in the gas phase gets perturbed when it is adsorbed on the TiO₂ solid and Kipreos et al. ⁵ identified 8 different hydroxyl groups in the 3000+ cm⁻¹ region. These hydroxyls show different behavior (change in intensity and wavenumber) with time as water is adsorbed. d) The 1620 cm⁻¹ is not likely from carbonate since only water

	was introduced in this experiment (Figure 4). Also, comparing our spectra in Figure S3 and S4 with results from the literature that studied water adsorption on TiO₂ (see reference ⁶) further confirms our assignment. The manuscript has been edited to briefly emphasize these points.
--	--

References

- (1) Liu, C.; Iyemperumal, S. K.; Deskins, N. A.; Li, G. Photocatalytic CO₂ Reduction by Highly Dispersed Cu Sites on TiO₂. *J. Photonics Energy* **2016**, *7* (1), 012004. <https://doi.org/10.1117/1.JPE.7.012004>.
- (2) Coloma, F.; Marquez, F.; Rochester, C. H.; Anderson, J. A. Determination of the Nature and Reactivity of Copper Sites in Cu-TiO₂ Catalysts. *Phys. Chem. Chem. Phys.* **2000**, *2* (22), 5320–5327. <https://doi.org/10.1039/b005331g>.
- (3) Van Every, K. W.; Griffiths, P. R. Characterization of Diffuse Reflectance FT-IR Spectrometry for Heterogeneous Catalyst Studies. *Appl. Spectrosc.* **1991**, *45* (3), 347–359.
- (4) Berman, A. Water Vapor in Vacuum Systems. *Vacuum* **1996**, *47* (4), 327–332. [https://doi.org/10.1016/0042-207X\(95\)00246-4](https://doi.org/10.1016/0042-207X(95)00246-4).
- (5) Kipreos, M. D.; Foster, M. Water Interactions on the Surface of 50 Nm Rutile TiO₂ Nanoparticles Using in Situ DRIFTS. *Surf. Sci.* **2018**, *677*, 1–7. <https://doi.org/10.1016/j.susc.2018.05.005>.
- (6) Green, I. X.; Tang, W.; Neurock, M.; Yates, J. T. Low-Temperature Catalytic H₂ Oxidation over Au Nanoparticle/TiO₂ Dual Perimeter Sites. *Angew. Chemie - Int. Ed.* **2011**, *50* (43), 10186–10189. <https://doi.org/10.1002/anie.201101612>.

REVIEWERS' COMMENTS:

Reviewer #1 (Remarks to the Author):

The authors have reasonably addressed the major concerns raised during the first round review by adding experimental data and extending discussion on the IR experiments. The main conclusions are now understandable and self-supportive, despite debates on the chemical nature of the proposed active sites remain further clarification in the future work. I have no more comments on this work.

Reviewer #2 (Remarks to the Author):

The authors have adequately addressed my previous comments. No further revisions are requested.